# Enantioselective organocatalytic synthesis of axially chiral aldehyde-containing styrenes via S$_N$Ar reaction-guided dynamic kinetic resolution

Fengyuan Guo[1,3], Siqiang Fang[1,3], Jiajia He[1,3], Zhishan Su[1] ✉ & Tianli Wang[1,2] ✉

The precise and efficient construction of axially chiral scaffolds, particularly toward the aryl-alkene atropoisomers with impeccably full enantiocontrol and highly structural diversity, remains greatly challenging. Herein, we disclose an organocatalytic asymmetric nucleophilic aromatic substitution (S$_N$Ar) reaction of aldehyde-substituted styrenes involving a dynamic kinetic resolution process via a hemiacetal intermediate, offering a novel and facile way to significant axial styrene scaffolds. Upon treatment of the aldehyde-containing styrenes bearing (o-hydroxyl)aryl unit with commonly available fluoroarenes in the presence of chiral peptide-phosphonium salts, the S$_N$Ar reaction via an exquisite bridged biaryl lactol intermediate undergoes smoothly to furnish a series of axially chiral aldehyde-containing styrenes decorated with various functionalities and bioactive fragments in high stereoselectivities (up to >99% ee) and complete E/Z selectivities. These resulting structural motifs are important building blocks for the preparation of diverse functionalized axial styrenes, which have great potential as efficient and privileged chiral ligands/catalysts in asymmetric synthesis.

The assembly of functionalized axially chiral compounds is not only the consistent theme in asymmetric catalysis but has also become the focus of cutting-edge research in the area of medicinal and material chemistry[1–4]. Among them, the frequent occurrence and importance of atropisomeric carbonyl scaffolds in various fields have received increasing attention in the synthetic community (Fig. 1a)[5,6]. An illustrative example is the axially chiral biaryl aldehyde catalysts that have been studied for their indispensable role in carbonyl catalysis, particularly offering an efficient strategy for direct asymmetric α-C−H functionalization of NH$_2$-unprotected primary amines, in which elegant contributions have been made by the groups of Guo[7,8] and Zhao et al.[9,10]. However, the lack of attention toward the skeletal diversity of

axially chiral biaryl aldehydes[11–21], let alone those acyclic axial styrene counterparts[22,23], is currently remarkable, which hampers the development of this field (Fig. 1b). Of note, the atroposelective synthesis of axially chiral acyclic styrenes remains an intrinsically formidable challenge due to the difficult control of E/Z selectivities and their relatively lower rotational barriers compared to either the cyclic aryl-alkene or the biaryl atropoisomers[24,25]. Recently, a breakthrough was made by Tan and co-workers, and they developed an organocatalytic approach towards atroposelectivly preparing axially chiral aldehyde-containing styrenes via a direct nucleophilic addition reaction of substituted diones with alkynals[22]. Shortly after, Shi and co-workers applied the Pd-catalyzed atroposelective C−H olefination to access this

[1]Key Laboratory of Green Chemistry & Technology of Ministry of Education, College of Chemistry, Sichuan University, Chengdu, PR China. [2]Beijing National Laboratory for Molecular Sciences, Beijing, China. [3]These authors contributed equally: Fengyuan Guo, Siqiang Fang, Jiajia He.
✉e-mail: suzhishan@scu.edu.cn; wangtl@scu.edu.cn

**Fig. 1 | Asymmetric catalytic synthesis of axially chiral aldehyde-containing styrenes. a** Representative axially chiral carbonyl natural products, pharmaceuticals, and catalysts. **b** Challenges towards axially chiral aldehyde compounds. **c** State-of-the-art developments of bridged biaryl (aza)-lactol system. **d** This work: asymmetric catalytic synthesis of axially chiral aldehyde-containing styrenes.

class of axially chiral styrenes[23]. Despite these impressive examples, designing structurally diverse and new axial styrenes with aldehyde units together with exploring a general yet powerful synthetic approach toward accessing such axially chiral molecules is still highly desirable, but challenging.

In 1995, Bringmann and co-workers disclosed an interesting bridged biaryl lactol system, which involved a ring-opening/ring-closing equilibrium between 1,1'-biaryl hydroxy aldehydes and axially prochiral biaryl O,O-acetal species, reasonably explaining the stereochemical leakage of the chiral biaryl hydroxy aldehyde in quantitatively chiral boron-induced reduction of biaryl lactones[26]. Later, an alternative bridged biaryl *aza*-lactol system was further developed and applied in the field of catalytic asymmetric transfer hydrogenation reactions by the groups of Akiyama[27] and Wang[28], which could enable the construction of axially chiral biaryl compounds in highly enantioselective fashion via a dynamic kinetic resolution (DKR) pathway (Fig. 1c). Quite recently, Chi and co-workers also described a related DKR strategy for the asymmetric cyanation reaction of racemic 2-arylbenzaldehydes catalyzed by stereochemically pure *N*-heterocyclic carbenes[29]. Inspired by these studies, we envisaged that a more efficient atroposelective synthetic strategy towards the assembly of structurally diversified axial aldehyde-containing styrenes could be

established via a dynamic kinetic transformation of (*o*-hydroxy)aryl-alkene aldehyde precursors.

Notably, this envisioned DKR strategy was actually based on the incorporation of appropriate fragments into the aryl-alkene hydroxy aldehyde precursors to intercept the dynamic kinetic interconversion between the hydroxy aldehydes and corresponding aryl-alkene O,O-acetals, and also to increase the steric hindrance enough to create high rotation barrier to maintain the integrity of the chiral axis. With this consideration in mind, the nucleophilic aromatic substitution (S$_N$Ar)[30] was thought to be the ideal reaction to reach this goal, because it is a conventional and efficient benchmark transformation for introducing an aromatic group onto anionic nucleophiles such as amines or phenols via the *ipso* substitution of arenes. While the development of catalytic asymmetric S$_N$Ar reactions has witnessed interesting yet sporadic advancements for the synthesis of enantioenriched triarylmethanes[31], α,α-disubstituted α-amino acids[32], cyclophanes[33] and axial biaryls[34], their applications in preparing more chiral counterparts currently remain underexplored.

In the past decades, phosphonium salt catalysis has become one of the central pillars in organocatalytic synthesis due to its huge potential and versatility for efficient assembly of diverse chiral molecules[35–39]. As a continuation of our interest in developing novel

and efficient asymmetric synthetic methodologies by chiral peptide-phosphonium salt (PPS) catalysis[40–42], we speculated that these highly tunable multifunctional ion-pair catalysts with extraordinary stereo-control ability would allow for the high reactivity and excellent atroposelectivity of our designed $S_NAr$-guided DKR process via a bridged aryl-alkene lactol system. To response the feasibility of this hypothesis, herein, we disclosed an unprecedented organocatalytic atroposelective synthesis of valuable axially chiral aldehyde-containing styrenes in high stereoselectivities and complete $E/Z$ selectivities via PPS-catalyzed $S_NAr$ reaction of (o-hydroxy)aryl-alkene aldehydes, wherein a dynamic kinetic resolution process by a ring-opening/ring-closing equilibrium between the aryl-alkene aldehydes and corresponding O,O-acetal intermediates was involved (Fig. 1d). Moreover, this method established a platform for rapid access to axially chiral styrene-based

derivatives such as carboxylic acids and dienes, thus offering an opportunity for developing new types of axially chiral aryl-alkene ligands/catalysts for asymmetric synthesis.

## Results
### Condition optimization

We initially commenced our investigation by the $S_NAr$ reaction of 1-enal substituted 2-naphthol **1a** with 2, 4-dinitrofluorobenzene **2a** (Fig. 2). Encouragingly, all of these bifunctional chiral phosphonium salt catalysts could promote this dynamic kinetic nucleophilic substitution reaction, providing the expected axially chiral products in nearly quantitative yields (99% yield). When the *L*-Val-derived phosphonium salts **P1**−**P4** were used, the reaction proceeded smoothly but with low ee values (entries 1−4). Notably, the phosphonium bromide

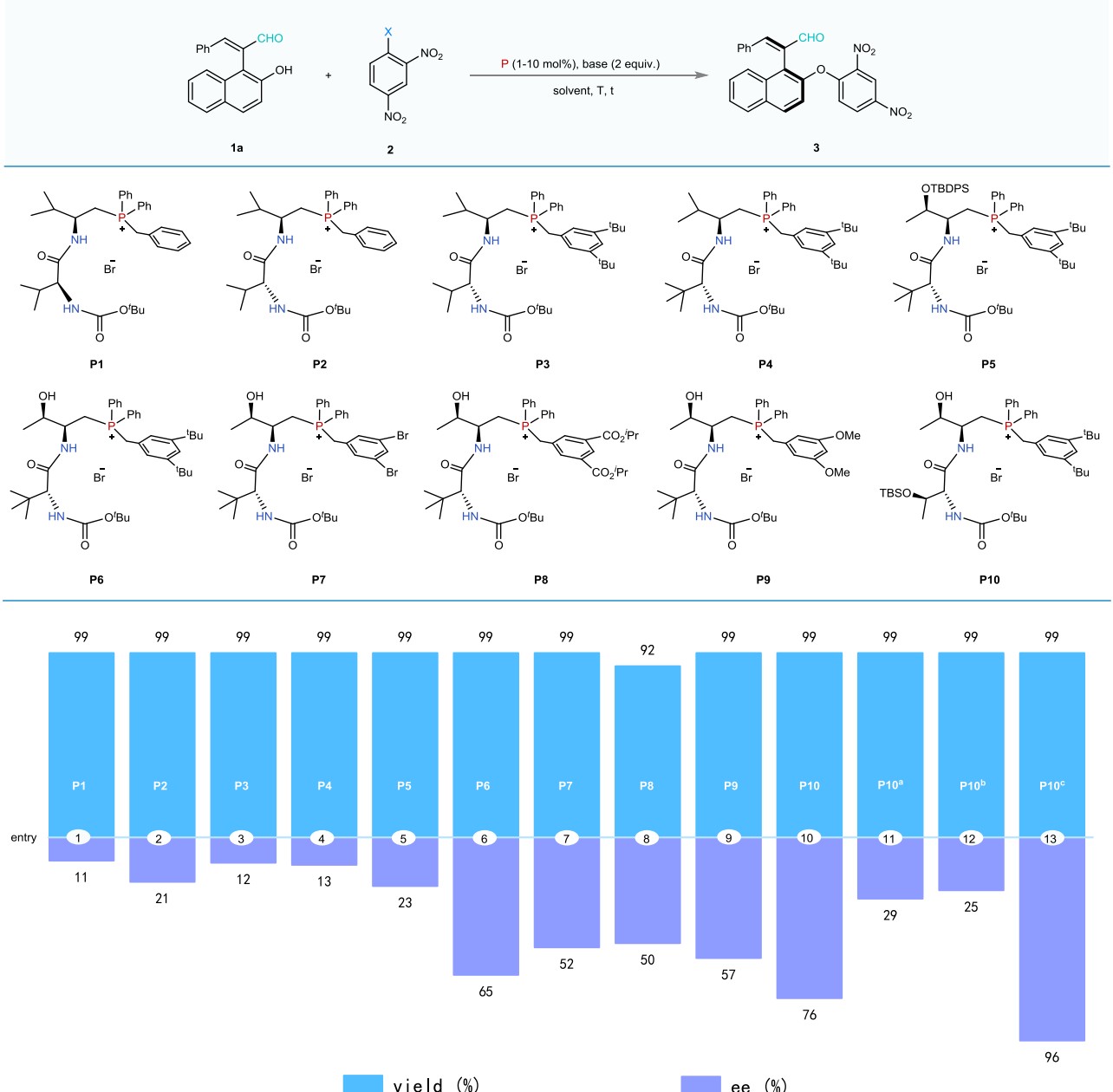

**Fig. 2 | Optimization of the reaction conditions.** Reaction conditions: **1a** (0.10 mmol), **2a** (X = F, 0.12 mmol), catalyst **P** (10 mol %) and $Cs_2CO_3$ (0.20 mmol) in toluene (1.0 ml) at rt; [a]With **2b** (X = Cl, 1-chloro-2, 4-dinitrobenzene) in above conditions; [b]With **2c** (X = Br, 1-bromo-2,4-dinitrobenzene) in above conditions; [c]With **P10** (1 mol %) and $K_2CO_3$ (0.20 mmol) in $CHCl_3$ (1.0 ml) at −10 °C for 24 h.

catalyst **P5** derived from *O*-TBDPS-*L*-threonine resulted in opposite enantioselectivity. On the other hand, when the phosphonium salt catalysts **P6**-**P9** were employed for this model reaction, the corresponding product was obtained with high yields (92–99% yield) in moderate ee values (entries 6–9). Inspired by these results, we further prepared the *L*-threonine-*O*-TBS-*D*-threonine-based phosphonium bromide **P10** and evaluated its catalytic reactivity and stereo-control capability in this S$_N$Ar reaction. As expected, the catalyst **P10** was found to be the best for promoting this reaction, affording the corresponding axially chiral product in quantitative isolated yield with good enantioselectivity (99% yield and 76% ee, entry 10). Then, we further optimized other reaction parameters such as bases and solvents. The combination of chloroform as solvent and 2.0 equivalents of K$_2$CO$_3$ as base could effectively improve the enantioselectivity (see Supporting Information for more details). Besides, the reaction temperature was also investigated. Lowering the temperature to −10 °C could lead to the desired axially chiral product in 96% ee with 99% yield (see Supporting Information for more details). Notably, while employing the 2, 4-dinitrochlorobenzene or 2, 4-dinitrobromobenzene instead of the 2, 4-dinitrofluorobenzene, we found that the reaction rate significantly slowed down, and the ee value was also substantially reduced (entries 11 and 12). To verify the catalytic efficiency and practicality of such PPS catalyst, we investigated the catalyst loading and found that the target product could be obtained with 99% yield and 96% ee by only employing 1 mol% of the catalyst **P10** (entry 13), demonstrating the high catalytic efficiency and practicality of this asymmetric synthetic protocol.

### Substrate scope

With the optimal reaction conditions in hand, we then focused on the generality of this protocol. At first, we explored a wide variety of (*o*-hydroxy)aryl-alkene aldehydes **1** installing different substituents on the aromatic rings Ar$^1$ and Ar$^2$, respectively. As illustrated in Fig. 3, the electronic properties and locations of the substituents on the aromatic ring (Ar$^1$) had no obvious effect on both isolated yields and enantioselectivities, providing the corresponding axial styrene products (**3**–**17**) in high yields and stereoselectivities. Of note, the aldehyde substrates **1** bearing heteroaromatic ring bonding with the C = C moiety were also suitable reaction partners, leading high reactivities and excellent enantioselectivities for the desired axially chiral styrene products (**18** and **19**). Then, the scope of aldehydes **1** bearing different substituent groups at another aryl ring (Ar$^2$) was also assessed. A diverse array of aromatic substituents was investigated, and all these aryl-substituted styrene aldehydes underwent this DK S$_N$Ar reaction with 2, 4-dinitrofluorobenzene (**2a**) smoothly, providing the corresponding axially chiral products (**20**–**30**) in excellent yields with high enantioselectivities. Of note, the quinolyl (Ar$^2$) substituted aldehyde was also suitable reaction partner for this DK S$_N$Ar transformation, thus leading to the axially chiral styrene aldehyde (**31**) in high yield and stereoselectivity. Subsequently, the universality of the nitro-fluorobenzene substrates **2** was also investigated, and the corresponding axially chiral styrene products (**32**, **33**) were generated in high yields with excellent asymmetric induction. More notably, when the 2, 4-dinitro-1, 5-difluorobenzene was employed for reacting with two equivalents of substrate **1a**, the chiral product containing two C(vinyl)-C(aryl) axes (**34**) was isolated in high yield (87% yield) with excellent enantioselectivity (>99% ee) and diastereoselectivity (>20:1 dr). Besides, the rotation barrier of axial styrene product (**3**) was also explored (see Supplementary Fig. 2 and Supplementary Fig. 3, Supplementary Table 5 and Supplementary Table 6 in Supporting Information for details) and the experimental result suggested that the rotation of compound **3** would be much difficult due to the high energy barrier of 26.5 kcal/mol. The absolute configurations of these atropisomeric styrene products were assigned based on the X-ray crystal structural analysis of chiral compound **3**.

### Synthetic applications

The wide substrate scope and excellent functional group compatibility of this dynamic kinetic S$_N$Ar reaction encourage us to explore its feasibility in the presence of substrates containing natural product fragments and/or biologically active moieties. Structurally complex 1-enal substituted 2-naphthols **1** with multiple functional groups derived from drug molecules and/or natural products are well-tolerated under the standard conditions, affording the corresponding products in high yields with excellent regioselectivities (**35**–**43**), which demonstrated a rich structural diversity for the axial products via this protocol (Fig. 4a). Furthermore, after the establishment of gram-scale synthesis of axially chiral styrene products with maintained reactivities and stereoselectivities, we proceeded to demonstrate their further synthetic utility (Fig. 4b). The Wittig reaction is the most common derivatization reaction for aldehydes, and thus we chose three Wittig reagents to react with the chiral product **17** under mild conditions. All three reactions afforded the *Z*-type products (**44**, **46**, and **47**) in high isolated yields with maintained enantioselectivities. In addition, under the sodium hydrosulfide conditions, the chiral styrene **44** could be easily transformed into important axial phenol compound **45** in excellent yield and enantioselectivity. Besides, the product **17** could be readily converted into other axially chiral styrene forms through reduction with sodium borohydride to yield the corresponding alcohol **48**, and via oxidation with the Pinnick oxidants to yield the axial carboxylic acid **49** in high yields and ee values, respectively.

### Mechanistic investigations

To gain preliminary insights into the reaction mechanism of this S$_N$Ar system, particularly regarding the reaction pathway and origins of asymmetric induction, a series of mechanistic experiments were carried out (Fig. 5). Firstly, we monitored the S$_N$Ar reaction between substrates **1a** and **2a** under the standard conditions to investigate the DKR process. As shown in Fig. 5a, it was discovered that the ee values of substrate **1a** constantly remained at 0, thus indicating that the substrate **1a** could occur racemization to meet the basic requirements of DKR under the standard condition. Initially, the ee values of product **3** exhibited an increasing trend, reaching a steady state at 6 hours (95% ee), which might be due to a strong background reaction at the early stage of this transformation. Furthermore, the linear relationship between the enantiopurities of chiral dipeptide-phosphonium salt catalyst **P10** and S$_N$Ar product **3** was determined. As illustrated in Fig. 5b, we applied the catalyst with a distinct enantiopurity value in the reactions under the standard conditions, respectively, and subsequently recorded the corresponding ee values of the product **3**. The resulting linear relationship, rather than a non-linear one, suggested that a monomeric phosphonium salt molecule might participate in the stereodetermining step. Then, the $^1$H NMR titration experiments were performed to gain further insights into the mechanism (Fig. 5c). In general, titration of the 1-enal a noticeable change in the substituted 2-naphthol **1a** to the optimal catalyst **P10** led to position of the OH and NH signals of the catalyst, while titration of 2, 4-dinitrofluorobenzene **2a** to this catalyst **P10** resulted in less change. As shown in Fig. 5d, we drew the Job plot curve, where the x-axis refers to the percentage of catalyst **P10**, and the y-axis refers to the interaction strength of **P10** with substrate **1a**.

The results suggested that a 1:1 binding pattern of **P10** with substrate **1a** was involved in this catalytic system. On the other hand, it was observed that catalyst **P10** significantly affected the position of the $^{19}$F NMR of substrate **2a**, indicating that there existed an interaction between the catalyst and the fluorine atom of substrate **2a**, which was also further confirmed by subsequent DFT calculations (see Supporting Information for more details). Besides, the OH/NH-blocked phosphonium salts **P10-1**, **P10-2**, and **P10-3** were prepared and applied to the model S$_N$Ar reaction. When the hydrogen-bonding site of the OH group in the *L*-threonine moiety of catalyst was blocked, the product **3**

**Fig. 3 | Substrate scope.** Unless other noticed, the reactions were performed with **1** (0.10 mmol), **2** (0.12 mmol), **P10** (1 mol%), K₂CO₃ (0.20 mmol) in CHCl₃ (1.0 ml) at −10 °C for 24 h. Isolated yield. The ee value was determined by HPLC analysis on a chiral stationary phase.

with the opposite configuration was obtained in lower enantioselectivity (29% ee). When both hydrogen-bonding sites (OH and NH) of the *L*-threonine moiety of catalyst were blocked, the enantioselectivity of product **3** obviously decreased. These mechanistic results suggested that all the hydrogen-bonding donor moieties of the catalyst played a crucial role for highly stereoselective control of this reaction.

To gain further insights into the origin of enantioselectivity in this $S_NAr$ reaction, density functional theory (DFT) calculations were performed using Gaussian 09 program at the M062X[43]-D3[44,45]/6-311 + G(d,p)(SMD, chloroform)//M062X-D3/6-31 G(d,p)(SMD, chloroform) level of theory. Four pathways associated with the formation of (*S*)-**3** and (*R*)-**3** were studied (see Supporting Information for more details). The energy profiles along two low-energy competing paths

(path I-*S/R*) were illustrated in Fig. 6. First, the catalyst **P10** activated the racemic (*R*)/(*S*)-**1a** anion and the substrate **2a** simultaneously, leading to the formation of **I-IM1-*S* ~ II-IM1-*R***. As shown in Fig. 6, **1a** was positioned by ion-pairing interaction (Supplementary Figure 10), while **2a** was activated through (N/O)H···O hydrogen-bonding. For **I-IM1-*S***, the (O) H₃···O₃ and (N)H₂···O₄ distances were 1.90 and 2.13 Å, respectively (Supplementary Figure 11). The corresponding Laplacian ($\nabla^2\rho$) on (3, -1) BCPs were 0.087 a.u. (d) and 0.055 a.u. (c) by Atoms-in-Molecules (AIM) analysis[46]. Due to strong H-bonding interaction ($E_{HB}$ = −5.73 and −2.83 kcal/mol)[47], **I-IM1-*S*** was more stable than the other three molecular complexes by 8.0–17.2 kcal/mol. Next, the C-O bonds in **I-IM2-*S* ~ II-IM2-*R*** were constructed via transition states **I-TS1-*S* ~ II-TS1-*R***, respectively. For **II-TS1-*S*** and **II-TS1-*R***, the F atom in **2a** was placed away

**Fig. 4 | Synthetic applications. a** Late-stage diversification. **b** Scale-up preparation and transformations of axial products.

from the OH group in the catalyst. Lack of stabilizing H-bonding interaction between OH group and F atom, **II-TS1-S** and **II-TS1-R** were less stable than **I-TS1-S** and **I-TS1-R** by 10.3 and 1.1 kcal/mol. For **I-TS1-S**, the (N)H$_1$···O$_1$ and (N)H$_2$···O$_4$ distances were 2.16 and 2.02 Å. AIM analysis indicated that the electron densities (ρ) at the (3, −1) bond critical points (BCPs) (a and e in Fig. 7) were 0.017 a.u. and 0.020 a.u., with the corresponding hydrogen bonding energies of $E_{HB(a)} = -3.05$ kcal/mol and $E_{HB(e)} = -3.72$ kcal/mol, respectively. As shown in Fig. 7, there existed

more stabilizing non-covalent interaction between the $^+$PPh$_2$CH$_2$Ar unit in **P10** and the naphthalene group in **1a**. Consequently, the ΔG of **I-TS1-S** was lower than that of **I-TS1-R** by 4.0 kcal/mol. In the final step, the **I-IM3-S** - **II-IM3-R** were formed, accompanying with the cleavage of C-F bond. With the aid of (O)H···F hydrogen-bonding, the energy barriers via **I-TS2-S** and **I-TS2-R** were significantly lower than those via **II-TS2-S** and **II-TS2-R**. In the view point of energy, the reaction preferred to occur along path I for leading to the major product with *S*-configuration.

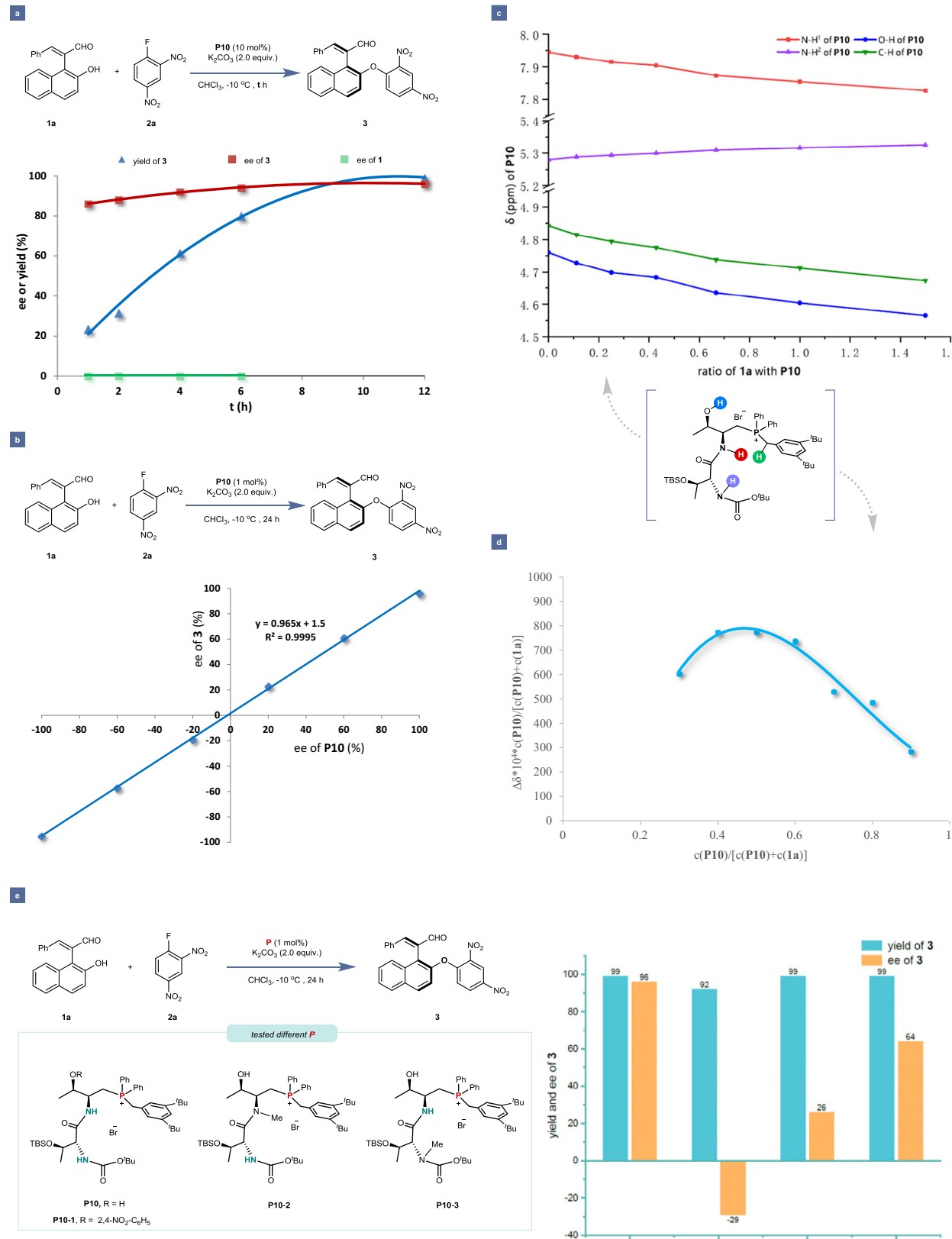

**Fig. 5 | Mechanistic investigations. a** Kinetic experiment towards a dynamic kinetic resolution (DKR) pathway. **b** Non-linear experiments. **c** [1]H NMR titration of catalyst **P10** with **1a**. **d** Job plot analysis. **e** Investigation of weak interaction of catalyst **P10**.

## Discussion

In summary, we have developed an atroposelective DKR reaction for the enantioselective preparation of axially chiral styrenes with aldehyde fragment. Racemic mixtures of 1-enal substituted 2-naphthols bearing

intrinsic axes and commercially available 2, 4-dinitrofluorobenzene derivatives were used as the reaction starting materials, together with employing a chiral peptide-phosphonium salt as the key organocatalyst. With this protocol, a great diversity of axially chiral styrenes with

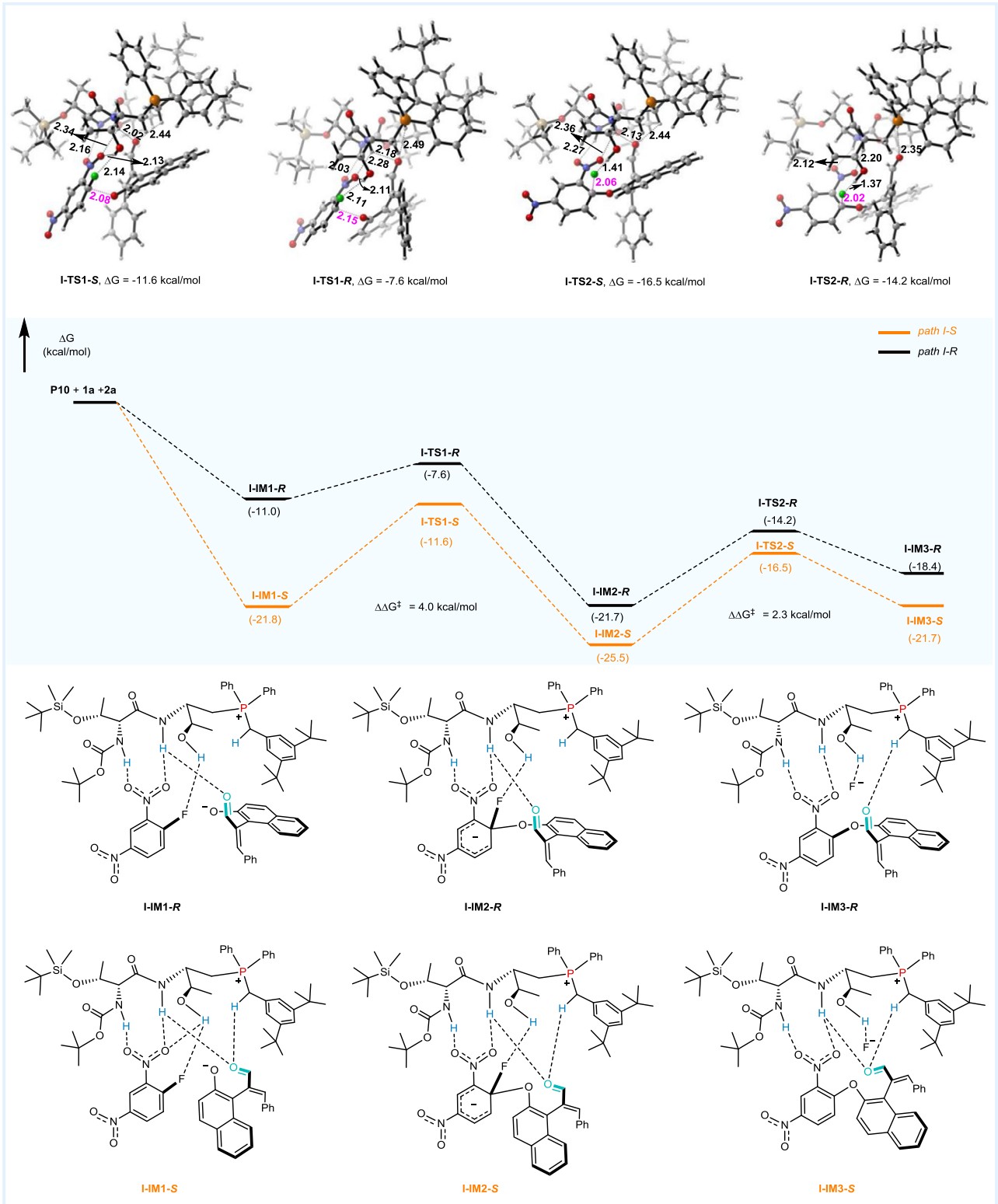

**Fig. 6 | Energy profiles along path I-*S* and I-*R*.** The relative free energies are given in kcal/mol.

aldehyde units, also bearing various substituents and substitution patterns, were easily prepared in high chemical yields with excellent enantioselectivities and complete *E/Z* selectivities under very low catalyst loading (1 mol%). The practicality and utility of this method were demonstrated by the gram-scale synthetic reactions and facile elaborations, particularly towards preparing novel axially chiral styrene molecules bearing an extended C═C unit and carboxylic acid, respectively,

which are an important family of atropisomeric building blocks that bear intrinsically synthetic challenge. Moreover, both experimental and computational mechanistic investigations revealed that the condensation of the phenol hydroxyl group with aldehyde unit of the racemic styrene substrates to form hemiacetal intermediates bearing five-membered ring was the key step for both rate- and stereo-determination. This DKR strategy guided by a $S_NAr$ reaction is expected to advance more

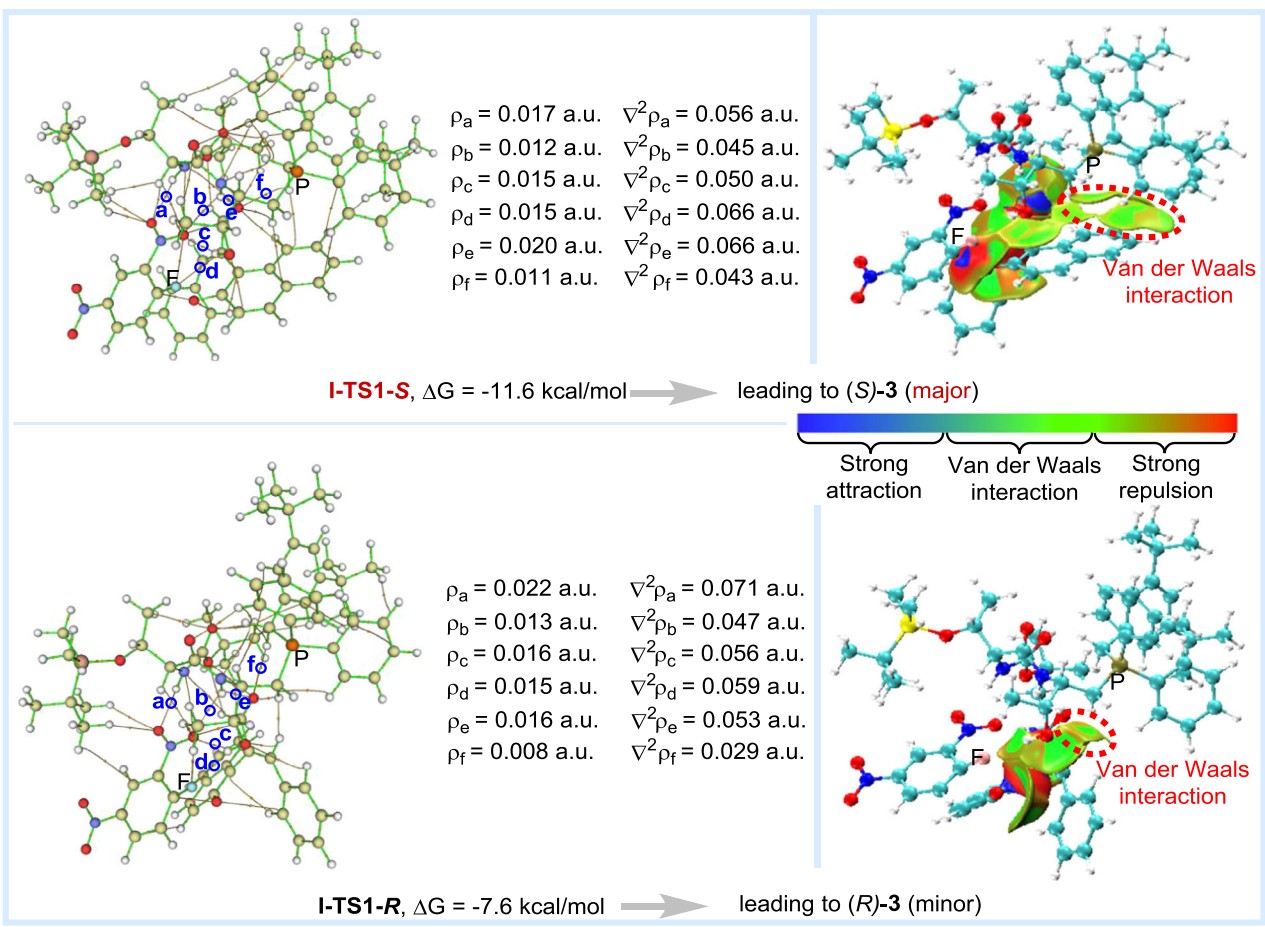

$\rho_a = 0.017$ a.u.   $\nabla^2\rho_a = 0.056$ a.u.
$\rho_b = 0.012$ a.u.   $\nabla^2\rho_b = 0.045$ a.u.
$\rho_c = 0.015$ a.u.   $\nabla^2\rho_c = 0.050$ a.u.
$\rho_d = 0.015$ a.u.   $\nabla^2\rho_d = 0.066$ a.u.
$\rho_e = 0.020$ a.u.   $\nabla^2\rho_e = 0.066$ a.u.
$\rho_f = 0.011$ a.u.   $\nabla^2\rho_f = 0.043$ a.u.

Van der Waals interaction

**I-TS1-*S***, $\Delta G$ = -11.6 kcal/mol ⟶ leading to (*S*)-**3** (major)

Strong attraction | Van der Waals interaction | Strong repulsion

$\rho_a = 0.022$ a.u.   $\nabla^2\rho_a = 0.071$ a.u.
$\rho_b = 0.013$ a.u.   $\nabla^2\rho_b = 0.047$ a.u.
$\rho_c = 0.016$ a.u.   $\nabla^2\rho_c = 0.056$ a.u.
$\rho_d = 0.015$ a.u.   $\nabla^2\rho_d = 0.059$ a.u.
$\rho_e = 0.016$ a.u.   $\nabla^2\rho_e = 0.053$ a.u.
$\rho_f = 0.008$ a.u.   $\nabla^2\rho_f = 0.029$ a.u.

Van der Waals interaction

**I-TS1-*R***, $\Delta G$ = -7.6 kcal/mol ⟶ leading to (*R*)-**3** (minor)

**Fig. 7 |** Laplacian ($\nabla^2\rho$) and electronic density ($\rho$) values of selected bond critical points (BCPs) in I-TS1-*S* and I-TS1-*R*, obtained by Atoms-in-Molecules (AIM) analysis. Independent gradient models of I-TS1-*S* and I-TS1-*R*, obtained by Multiwfn and VMD software.

explorations of the related organic synthesis, particularly toward creating diverse axially chiral molecules for new chiral ligand/catalyst discovery, and these studies are currently ongoing in our laboratory.

## Methods
### General procedure for the catalytic asymmetric synthesis of axially chiral styrenes 3-43
To a dried round-bottom flask with a magnetic stirring bar were added substrates 1 (0.10 mmol) and 2 (0.12 mmol), followed by the addition of $K_2CO_3$ (27.6 mg, 0.2 mmol), catalyst P10 (0.9 mg, 0.001 mmol) and the solvent $CHCl_3$ (1.0 mL). The reaction mixture was stirred at −10 °C for 24 h and then directly subjected to column chromatography on silica gel (5:1 DCM/petroleum ether) to give the desired pure products in 87−99% isolated yields.

## Data availability
Crystallographic data for the structures reported in this Article have been deposited at the Cambridge Crystallographic Data Centre, under deposition number CCDC 2162443 (**3**). Copies of the data can be obtained free of charge via https://www.ccdc.cam.ac.uk/structures/. Data related to materials and methods, optimization of conditions, experimental procedures, mechanistic experiments, and spectra are provided in the Supplementary Information. All data are available from the corresponding authors upon request.

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

## Acknowledgements
We acknowledge financial support from the National Natural Science Foundation of China (22222109, 21971165, and 21921002 for T.W.), the National Key R&D Program of China (2018YFA0903500 for T.W.), the Sichuan Science Foundation for Distinguished Young Scholars (2023NSFSC1921 for T.W.), Beijing National Laboratory for Molecular Sciences (BNLMS202101 for T.W.), Fundamental Research Funds from Sichuan University (2020SCUNL108 for T.W.), and Fundamental Research Funds for the Central Universities. We also acknowledge the College of Chemistry and the Analytical & Testing Center of Sichuan University, and particularly we thank Dr. Jing Li and Dr. Dongyan Deng from the College of Chemistry Sichuan University for HRMS and NMR testing, respectively.

## Author contributions
T.W. conceived and designed the study, and wrote the paper. F.G. and S.F. jointly conducted the experiments described in this manuscript and analyzed the data. F.G. also conducted the NMR and HPLC studies together with the crystallographic studies. J.H. and Z.S. performed the DFT studies. All the authors contributed to the analysis and interpretation of the data.

## Competing interests
The authors declare no competing interests.
