## [Peer Review File · Nature Communications]

REVIEWER COMMENTS

Reviewer #1 (Remarks to the Author):

In this manuscript, Wang and coworkers presents a novel and highly efficient catalytic atroposelective synthesis of axially chiral aldehyde-containing styrenes via SNAr reaction-guided dynamic kinetic resolution. Various aspects of this study have been executed proficiently. Firstly, the innovative approach, which has not been previously exploited for this purpose, deserves commendation for its conceptual basis. The well-conducted catalyst selection and screening led to the discovery of an excellent catalyst among the various phosphonium salts presented in Table 1. The scope exemplification is also almost acceptable, with numerous intriguing issues tested in which the results were excellent. Moreover, several valuable mechanistic studies are also provided, including investigations of configurational stability that appear to be well-executed. Kinetic investigations, NMR titration studies, nonlinear effect studies and systematic DFT calculations are all encompassed within this research. In conclusion, this reviewer recommends that this manuscript can be accepted for publication in Nature Communications after addressing the following minor issues.

1) In page 7, Figure 3B, the Wittig reagent which formed product 47 should be $\text{Ph}_3\text{P}=\text{CHCO}_2\text{tBu}$.

2) The Figure 4D (Job plot analysis) is not mentioned. An explanation should be included in the manuscript.

3) Has the author attempted using ketones as substrates in this DKR approach, and what were the catalytic results? If these reactions are feasible, this reviewer recommends that the author include these findings in the manuscript.

4) A related review should be cited: Recent advances in catalytic asymmetric construction of atropisomers. Chem. Rev. 2021, 121, 4805.

Reviewer #3 (Remarks to the Author):

In this excellent work Wang, Su and co-workers describes an efficient protocol for catalytic atroposelective synthesis of a series of axially chiral aldehyde-containing styrenes through catalytic dynamic kinetic resolution (DKR) relying on a “bridged biaryl lactol” enabled racemization. The strategy also relies on the well-designed chiral phosphonium salts, which have been extensively studied in the authors group. The merit of this work also includes the overall idea by utilizing SNAr reaction to terminate the atropoactive phenol anion. The manuscript clearly presented the catalyst screening, substrate scope investigation. Generally, good yields and excellent enantioselectivity were achieved. The studies of configurational stability are fine and appear to be well done, securing the stability of the final products. Moreover, mechanistic studies were well performed, which is well welcome for synthetic methodology development. Kinetics, NMR titration studies, nonlinear effect

studies, DFT calculations - it's all there. All in all, this is excellent paper and I would strongly support this work publication in Nat. Commun. after consideration of the following comments:

(1) A minor omission seems to be the lack of exploration of biaryl substrates via this “bridged biaryl lactol” racemization strategy. Did the author try? How about the catalytic asymmetric results?

(2) The full language polishing and revision of the manuscript is needed, such as spelling mistakes which is not suitable for final publication.

(3) It seems the structure of compound 34 (two configurations of two axes are different) in the Supporting Information is wrong. The picture of compound 34 in SI is a symmetric one.

Overall, the manuscript presents valuable insights into the reaction mechanism and origins of asymmetric induction. However, there are several points that need clarification and additional information. The following sections outline specific issues identified within the manuscript and suggest potential solutions for improvement.

- The authors state that a series of mechanistic experiments were conducted to gain insights into the reaction mechanism, particularly regarding the reaction pathway and origins of asymmetric induction. However, it is argued that a different approach, where theoretical studies are conducted first, would be more advantageous. In recent years, the trend has shifted towards conducting theoretical studies before proceeding with extensive experimental testing. This approach allows researchers to gain initial mechanistic insights, design the best scope of substrates and catalysts, and reduce the time and resources spent on unsuccessful laboratory testing. By understanding the reaction mechanism through theoretical studies, researchers can make informed decisions and optimize experimental conditions, leading to more efficient and successful experimental outcomes. Considering the aforementioned counterargument, it is recommended that the authors re-evaluate their approach and consider conducting theoretical studies before experimental testing for further studies. This approach will provide a solid foundation for designing experimental procedures and optimizing reaction conditions, ultimately improving the overall efficiency of the study.
- The manuscript mentions four different energy profiles, but the authors explored the same binding mode but with different orientations. However, the nomenclature used for these binding modes (I to IV) is misleading. A clarification is needed, same binding mode, orientation A, and within that particular orientation, enantiomer *R* and *S*, for instance. Otherwise it seems that the authors have explored the whole conformational space, which unfortunately is not true in this case. The manuscript should address whether the authors explored different binding modes and provide a clear explanation of how they determined that the substrate binds in the presented manner. Conformational analysis should be performed and included to support the proposed binding modes.

Besides, it is clear that in orientation B, the negatively charged oxygen is really far from the C-F to attack, why then study that particular pathway?

- It would be very interesting to see the penalty from the pre-TS to the TS in terms of geometry parameters, as well as the HBs for *R* and *S* in orientation A.

- The manuscript depicts different hydrogen bonds in the binding mode on the left side of Figure 6. However, these hydrogen bonds are not characterized in terms of distance and angles in the figure, it would be very beneficial, instead of the geometry, add the molecular graph of AIM with the main BCPs and the corresponding density values and discuss them within the text. The manuscript should also elaborate on the QTAIM outcome mentioned for SI and provide a clearer explanation within the main manuscript. Besides, why on the left those important hydrogen bonds do not appear in the NCI plots?

- The manuscript mentions the values of EHB (strong H-bonding interaction) without specifying the source of these values. It is essential to provide clarity on how these values were calculated, whether using AIM or other methods. Additionally, the manuscript should address the characterization of non-covalent interactions (NCIs) between the naphthalene group in 1a and the Ar group in P10. If these interactions have been characterized using AIM, the outcome should be clearly presented and discussed in detail. Characterisation of those interactions is vital.
- The manuscript mentions an "ion-pairing interaction" but does not provide sufficient evidence or proof for this claim. It is suggested to include supporting evidence, such as NBO analysis or charge calculations, to substantiate the ion-pairing interaction. Where are the charges located? It seems more likely to have a hydrogen bond assisted ion-pair.
- Additionally, the computational details regarding the SMD solvent model should be clarified, specifying whether it was applied at the single-point level or during the optimisation process too.

- In the different free energy profiles the entrance channel is the pre-TS, however, from the theoretical perspective, the entrance channel is described as the energy of all the different components isolated. The energy profiles must be changed accordingly.

In conclusion, the manuscript "Enantioselective Organocatalytic Synthesis of Axially Chiral Aldehyde-containing Styrenes via SNAr Reaction-Guided Dynamic Kinetic Resolution" provides valuable insights into the reaction mechanism and origins of asymmetric induction. However, there are concerns regarding the sequencing of experimental and computational results, nomenclature, characterization of hydrogen bonds and NCIs, explanation of interaction energies, ion-pairing interactions, computational details. Addressing these concerns will greatly enhance the clarity, reliability, and overall quality of the manuscript. I recommend the authors revise and provide further information to address the mentioned areas for improvement.

July 10, 2023

Dear Reviewers,

Thank you for your valuable suggestions and comments. According to your comments, we have carried out additional experiments and thereby revised the manuscript. Detailed replies are listed below.

Reviewer 1:

Comment 1: In page 7, Figure 3B, the Wittig reagent which formed product 47 should be $\text{Ph}_3\text{P}=\text{CHCO}_2^t\text{Bu}$.

Response: We appreciate this reviewer for this kind reminder. We have addressed this issue by making the necessary corrections.

Comment 2: The Figure 4D (Job plot analysis) is not mentioned. An explanation should be included in the manuscript.

Response: We appreciate this reviewer for this kind reminder. We have added the corresponding explanation of Figure 4D (Job plot analysis) in the revised manuscript.

Comment 3: Has the author attempted using ketones as substrates in this DKR approach, and what were the catalytic results? If these reactions are feasible, this reviewer recommends that the author include these findings in the manuscript.

Response: We appreciate the reviewer for this valuable suggestion. We have attempted to use ketones as substrates; however, we did not observe this DKR process (as shown in the following, 0% ee of product, no DKR process). We hypothesize that this may be attributed to the lower reactivity of ketones compared to aldehydes, and thus the formation of hemiketals from ketones is more challenging than that of hemiacetals from aldehydes. Therefore, it is difficult to achieve racemization of ketone substrates under the similar alkaline conditions.

Comment 4: A related review should be cited: Recent advances in catalytic asymmetric construction of atropisomers. *Chem. Rev.* **2021**, *121*, 4805.

Response: We appreciate the reviewer for this valuable suggestion. We have included this review in the list of references cited in the manuscript (as ref. 4 in this revised manuscript).

Reviewer 2:

We appreciate the reviewer's constructive comments and suggestions. We wish to provide our responses and the corresponding modifications made in the manuscript in the following aspects accordingly.

Comment 1: The authors state that a series of mechanistic experiments were conducted to gain insights into the reaction mechanism, particularly regarding the reaction pathway and origins of asymmetric induction. However, it is argued that a different approach, where theoretical studies are conducted first, would be more advantageous. In recent years, the trend has shifted towards conducting theoretical studies before proceeding with extensive experimental testing. This approach allows researchers to gain initial mechanistic insights, design the best scope of substrates and catalysts, and reduce the time and resources spent on unsuccessful laboratory testing. By understanding the reaction mechanism through theoretical studies, researchers can make informed decisions and optimize experimental conditions, leading to more efficient and successful experimental outcomes. Considering the aforementioned counterargument, it is recommended that the authors re-evaluate their approach and consider conducting theoretical studies before experimental testing for further studies. This approach will provide a solid foundation for designing experimental procedures and optimizing reaction conditions, ultimately improving the overall efficiency of the study.

Response: Thank you very much! We agree with your point well that theoretical studies before proceeding with extensive experimental testing would be helpful to realize more efficient and successful experimental outcomes by reducing the time and resources spent on unsuccessful laboratory testing. However, this reaction system is complicated, involving multiple weak-bond interactions (i.e. hydrogen-bonding and ion-pairing interactions) that contributed greatly to the stereoselectivities. So, we performed mechanistic experiments ahead of DFT calculations. The results of H-bond titration experiments and other some control experiments indeed help us to determine the possible binding sites in the catalyst, and proposed a hydrogen-bond assisted ion-pair model for further theoretical simulation. Thank you very much for your guidance, and we will perform more theoretically guided research in our future studies.

Comment 2: The manuscript mentions four different energy profiles, but the authors explored the same binding mode but with different orientations. However, the nomenclature used for these binding modes (I to IV) is misleading. A clarification is needed, same binding mode, orientation A, and within that particular orientation, enantiomer *R* and *S*, for instance. Otherwise it seems than the authors have explored the whole conformational space, which unfortunately is not true in this case. The manuscript should address whether the authors explored different binding modes and provide a clear explanation of how they determined that the substrate binds in the

presented manner. Conformational analysis should be performed and included to support the proposed binding modes.

Besides, it is clear that in orientation B, the negatively charged oxygen is really far from the C-F to attack, why then study that particular pathway?

Response: Thanks for this kind and important suggestion. According to your suggestion, four paths (I-IV) were re-named to I-S, I-R, II-S and II-R, respectively. For I-S and I-R in Orientation A you mentioned, the F atom in the substrate **2a** was positioned by the OH group. For II-S and II-R, the F atom in **2a** was placed away from the OH group. We have modified the relevant parts in the main text and Supplementary Information.

In the DFT calculations, we first carried out conformational search for catalyst **P10**. Molecular dynamics (MD) simulations with xTB 6.3 code and GFN0-xTB method were performed to sample the conformational space of catalyst **P10**. Then, Molclus software with GFN2-xTB method was used to obtain 54 candidate structures (criteria: energy difference < 0.5 kcal/mol and RMSD < 0.5 Å). In the last step, four low-energy structures were further optimized using Gaussian 09 program at the M062X-D3/6-31G(d,p)(SMD, chloroform) level of theory, and the lowest-energy conformer was used for theoretical simulation in the catalytic reaction.

Based on H-bonding titration experiment and our previous work, a catalyst-substrate interaction model was proposed. The NH and OH groups of the catalyst **P10** activated 2, 4-dinitrofluorobenzene (**2a**) by hydrogen bonding, and the positively charged phosphonium cation stabilized anionic **1a** by ion-pairing interaction. We found that the bulky substituents (e.g. OTBS) and H-bonds raised by the OH group in the catalyst limited conformational change of the two substrates in the intermediates or transition states. Accordingly, these reactive species became less flexible. In the structural optimization of TSs, we also considered some possible binding modes (for example, two NH groups in the catalyst interacted with the NO₂ group of **2a**, and the OH group simultaneously interacted with O atom of **1a** by H-bond). The transition states with the lowest energy were used in the manuscript.

For comparison, we also considered the pathways involving orientation B for the two substrates in the DFT calculations. As you mentioned, the negatively charged oxygen in the substrate **1a** is really far from the C-F bond to attack. As expected, the pathways via orientation B are unfavorable kinetically.

Comment 3: It would be very interesting to see the penalty from the pre-TS to the TS in terms of geometry parameters, as well as the HBs for *R* and *S* in orientation A.

Response: Thank you very much! According to your suggestion, we have added the optimized geometries of pre-TS (i.e., I-IM1-*S* and I-IM1-*R*) in the Figure S11 (as shown the following). Additionally, AIM analysis of I-IM1-*S* and I-IM1-*R* are also included.

Supplementary Figure S11. Optimized structures of I-IM1-*S* and I-IM1-*R*. The distances are in Å. Laplacian ($\nabla^2 \rho$) and electron density (ρ) values of selected bond critical points (BCPs) in I-IM1-*S* and I-IM1-*R*, obtained by AIM analysis.

Comment 4: The manuscript depicts different hydrogen bonds in the binding mode on the left side of Figure 6. However, these hydrogen bonds are not characterized in terms of distance and angles in the figure, it would be very beneficial, instead of the geometry, add the molecular graph of AIM with the main BCPs and the corresponding density values and discuss them within the text. The manuscript should also elaborate on the QTAIM outcome mentioned for SI and provide a clearer explanation within the main manuscript. Besides, why on the left those important hydrogen bonds do not appear in the NCI plots?

Response: Thanks for this valuable suggestion. According to your suggestion, we have performed Atoms-in-Molecules (AIM) analysis for transition state I-TS1-*S* and I-TS1-*R* by using Multiwfn 3.8 (dev) software. The positive Laplacian ($\nabla^2 \rho$) on (3, -1) BCPs indicated that there exists hydrogen bonding between the catalyst and the two substrates. The electron density (ρ) and the corresponding Laplacian ($\nabla^2 \rho$) on (3, -1) BCPs in I-TS1-*S* and I-TS1-*R* have been shown in Figure 6 in this revised version of manuscript. Moreover, the discussion about the results of AIM has also been added in the main text (see page 11), which was highlighted in yellow background. For NCI plots in Figure 6, we just visualized the interaction between the cavity around $^+PPh_2CH_2Ar$ moiety and substrate. Besides, we have added the whole NCI plots in Figure S12 in this revised Supplementary Information (as shown in the following figure).

Supplementary Figure S12. Optimized structures of I-TS1-S and I-TS1-R. The distances are in Å. Laplacian ($\nabla^2 \rho$) and electron density (ρ) values of selected bond critical points (BCPs) in I-TS1-S and I-TS1-R, obtained by AIM analysis. Non-covalent interaction (NCI) plots for I-TS1-S and I-TS1-R, obtained by Multiwfn and VMD softwares.

Comment 5: The manuscript mentions the values of EHB (strong H-bonding interaction) without specifying the source of these values. It is essential to provide clarity on how these values were calculated, whether using AIM or other methods. Additionally, the manuscript should address the characterization of non-covalent interactions (NCIs) between the naphthalene group in **1a** and the Ar group in **P10**. If these interactions have been characterized using AIM, the outcome should be clearly presented and discussed in detail. Characterisation of those interactions is vital.

Response: Thank you very much! We calculated the hydrogen bonding energy by using the formula of $E_{\text{HB}} = -223.08 \times \rho(\text{BCP}) + 0.7423$ (neutral hydrogen bonding system, kcal/mol) (Ref. 48: *J. Comput. Chem.* **2019**, *40*, 2868-2881). The electron density (ρ) at the (3, -1) bond critical points (BCP) was obtained by AIM analysis, using Multiwfn 3.8 (dev) software. We also analyzed the characters of non-covalent interactions (NCIs) between the naphthalene group in **1a** and the Ar group in **P10** in I-TS1-S. More stabilizing non-covalent interaction was observed between PPh₂ unit and naphthalene group in **1a** substrate. For I-TS1-R, the naphthalene group in **1a** was far away from the Ar group in **P10**, and there was not non-covalent interaction between them. We optimized the Figure 6, and also added the relevant discussion in this revised text (see page 11).

Fig. 6 Laplacian ($\nabla^2\rho$) and electronic density (ρ) values of selected bond critical points (BCPs) in I-TS1-S and I-TS1-R, obtained by Atoms-in-Molecules (AIM) analysis. Independent gradient models of I-TS1-S and I-TS1-R, obtained by Multiwfn and VMD softwares.

Comment 6: The manuscript mentions an "ion-pairing interaction" but does not provide sufficient evidence or proof for this claim. It is suggested to include supporting evidence, such as NBO analysis or charge calculations, to substantiate the ion-pairing interaction. Where are the charges located? It seems more likely to have a hydrogen bond assisted ion-pair.

Response: Thank you very much! According to your suggestion, we performed NBO analysis for the two TSs. The corresponding charge populations were shown in Figure

S10 (as shown in the following). The calculations indicated that the charges accumulated on the P atom of cation catalyst **P10** were 1.637 for I-IM1-S and 1.640 for I-IM1-R, respectively. The total charges of **1a** anion moiety were -0.947 for I-IM1-S and -0.930 for I-IM1-R, with the charges on the O atoms of **1a** anion of -0.624, -0.896, -0.667 and -0.802, respectively. Thus, we considered that there existed strong ion-pairing interaction between the cation catalyst **P10** and **1a** anion substrate.

Figure S10. Charge population in I-IM1-S and I-IM1-R, obtained by natural bond orbital (NBO) analysis.

Comment 7: Additionally, the computational details regarding the SMD solvent model should be clarified, specifying whether it was applied at the single-point level or during the optimisation process too.

Response: Thank you very much! The SMD solvent model was used for structural optimization as well as the single-point energy calculation.

Comment 8: In the different free energy profiles the entrance channel is the pre-TS, however, from the theoretical perspective, the entrance channel is described as the energy of all the different components isolated. The energy profiles must be changed accordingly.

Response: Thank you very much! According to your suggestion, we revised the free energy profiles using the sum of the energy of the catalyst and two substrates as reference in energy. The free energy profiles in manuscript and SI have been updated.

Reviewer 3:

Comment 1: A minor omission seems to be the lack of exploration of biaryl substrates via this “bridged biaryl lactol” racemization strategy. Did the author try? How about the catalytic asymmetric results?

Response: We appreciate the reviewer for this valuable suggestion. We have also investigated the use of bridged biaryl lactol (Of note, this type of compound exists in the form of a hemiacetal, which was also observed in previous reports) as a substrate, and finally found that it could also be subjected to this racemization strategy to yield the corresponding bridged biaryl axially chiral aldehydes, although only moderate ee

value (38% ee) was obtained without further optimization of the reaction conditions (as shown in the following).

Comment 2: The full language polishing and revision of the manuscript is needed, such as spelling mistakes which is not suitable for final publication.

Response: We appreciate this reviewer for this kind reminder. We have conducted a comprehensive review of the manuscript, and thus addressed some spelling errors and refined the language to improve its readability and clarity.

Comment 3: It seems the structure of compound **34** (two configurations of two axes are different) in the Supporting Information is wrong. The picture of compound **34** in SI is a symmetric one.

Response: We appreciate this reviewer for this kind reminder. We have revised the structure of compound **34** in the Supporting Information.

Yours sincerely,

Prof. Dr. Tianli Wang

Key Laboratory of Green Chemistry & Technology of Ministry of Education, College of Chemistry, Sichuan University, 29 Wangjiang Road, Chengdu 610064 (P. R. China)

E-mail: wangtl@scu.edu.cn

REVIEWERS' COMMENTS

Reviewer #2 (Remarks to the Author):

The authors have made significant improvements in response to the reviewer's comments, and the manuscript has benefited greatly from these revisions. I am pleased to say that the majority of the concerns have been effectively addressed. However, there is one remaining issue that requires further clarification.

The authors were asked to provide more information regarding the characterization of the suggested "strong ion-pair" upon complexation. While the revisions made by the authors are commendable, I believe that the statement implying the presence of a "strong ion-pair" should be revised due to a specific reason. The charges within the complex alone do not serve as conclusive proof of an ion-pair, especially when the atoms involved are positioned significantly far apart.

To rectify this, I strongly recommend that the authors revise the statement to refer to an "ion-pair" without the descriptor "strong." This adjustment will appropriately reflect the spatial distance between the two atoms within the complex.

Therefore, based on the improvements made, I recommend that this revised version of the manuscript be accepted for publication pending the authors' revision of the statement as suggested above.

Reviewer #2 (Remarks to the Author):

Comment 1: The authors have made significant improvements in response to the reviewer's comments, and the manuscript has benefited greatly from these revisions. I am pleased to say that the majority of the concerns have been effectively addressed. However, there is one remaining issue that requires further clarification.

The authors were asked to provide more information regarding the characterization of the suggested "strong ion-pair" upon complexation. While the revisions made by the authors are commendable, I believe that the statement implying the presence of a "strong ion-pair" should be revised due to a specific reason. The charges within the complex alone do not serve as conclusive proof of an ion-pair, especially when the atoms involved are positioned significantly far apart.

To rectify this, I strongly recommend that the authors revise the statement to refer to an "ion-pair" without the descriptor "strong." This adjustment will appropriately reflect the spatial distance between the two atoms within the complex.

Therefore, based on the improvements made, I recommend that this revised version of the manuscript be accepted for publication pending the authors' revision of the statement as suggested above.

Response: We sincerely appreciate your detailed and constructive comments on our manuscript. According to your suggestion, the response to **Comment 6** in the previous version, "there existed strong ion-pairing interaction between the cation catalyst **P10** and **1a** anion substrate"), was corrected to "there existed ion-pairing interaction between the cation catalyst **P10** and **1a** anion substrate". In the main text, we used the statement of "**1a** was positioned by ion-pairing interaction (Supplementary Figure 10)".